# Inflammation Status and Body Composition Predict Two-Year Mortality of Patients with Locally Advanced Head and Neck Squamous Cell Carcinoma under Provision of Recommended Energy Intake during Concurrent Chemoradiotherapy

**DOI:** 10.3390/biomedicines10020388

**Published:** 2022-02-06

**Authors:** Yu-Ching Lin, Cheng-Hsu Wang, Hang Huong Ling, Yi-Ping Pan, Pei-Hung Chang, Wen-Chi Chou, Fang-Ping Chen, Kun-Yun Yeh

**Affiliations:** 1Department of Medical Imaging and Intervention, Chang Gung Memorial Hospital, College of Medicine, Keelung & Chang Gung University, Taoyuan 333007, Taiwan; yuching1221@cgmh.org.tw; 2Osteoporosis Prevention and Treatment Center, Chang Gung Memorial Hospital, Keelung 20401, Taiwan; fangping@cgmh.org.tw; 3Division of Hemato-Oncology, Department of Internal Medicine, Chang Gung Memorial Hospital, College of Medicine, Keelung & Chang Gung University, Taoyuan 333007, Taiwan; chwang@cgmh.org.tw (C.-H.W.); xianfang87@cgmh.org.tw (H.H.L.); Ich2682@cgmh.org.tw (P.-H.C.); 4Department of Nutrition, Chang Gung Memorial Hospital, Keelung 20401, Taiwan; yipyng@cgmh.org.tw; 5Division of Hemato-Oncology, Department of Internal Medicine, Chang Gung Memorial Hospital, College of Medicine, Linkou & Chang Gung University, Taoyuan 333007, Taiwan; f12986@cgmh.org.tw; 6Department of Obstetrics and Gynecology, Chang Gung Memorial Hospital, Keelung 20401, Taiwan; 7Healthy Aging Research Center, College of Medicine, Chang Gung University, Taoyuan 333007, Taiwan

**Keywords:** head and neck cancer, concurrent chemoradiotherapy, calorie supply, mortality, body composition, inflammation, nutrition

## Abstract

Only few prospective cohort trials have evaluated the risk factors for the 2-year mortality rate between two patient subgroups with locally advanced head and neck squamous cell carcinoma (LAHNSCC): oral cavity cancer with adjuvant concurrent chemoradiotherapy (CCRT) (OCC) and non-oral cavity cancer with primary CCRT (NOCC), under the recommended calorie intake and investigated the interplay among calorie supply, nutrition–inflammation biomarkers (NIBs), and total body composition change (TBC), as assessed using dual-energy X-ray absorptiometry (DXA). Patients with LAHNSCC who consumed at least 25 kcal/kg/day during CCRT were prospectively recruited. Clinicopathological variables, blood NIBs, CCRT-related factors, and TBC data before and after treatment were collected. Factor analysis was performed to reduce the number of anthropometric and DXA-derived measurements. Cox proportional hazards models were used for analysis. We enrolled 123 patients with LAHNSCC (69 with OCC and 54 with NOCC). The mean daily calorie intake correlated with the treatment interval changes in total body muscle and fat. Patients consuming ≥30 kcal/kg/day had lower pretreatment levels but exhibited fewer treatment interval changes in anthropometric and DXA measurements than patients consuming <30 kcal/kg/day. In the multivariate analysis of the 2-year mortality rate, the prognostic influence of the recommended calorie intake could not be confirmed, but different risk factors (performance status, pretreatment platelet-to-lymphocyte ratio, and treatment interval body muscle changes in patients with OCC; age, pretreatment neutrophil-to-lymphocyte ratio, and body fat storage in patients with NOCC) showed independent effects. Therefore, the inflammation status and body composition, but not the recommended calorie supply, contribute to the 2-year mortality rate for patients with LAHNSCC receiving CCRT.

## 1. Introduction

Most patients with locally advanced head and neck squamous cell carcinoma (LAHNSCC) require concurrent chemoradiotherapy (CCRT) as either adjuvant therapy following surgery for patients with oral cavity cancer (OCC), or curative-intent primary CCRT therapy for patients with non-oral cavity cancer (NOCC, pharynx, larynx, and paranasal sinus), to improve disease control [1,2]. Nonetheless, a high prevalence of malnutrition at the time of diagnosis owing to the patients’ inherent characteristics and tumor factors, resulting in a deteriorating systemic metabolic response, or the inevitable progression of malnutrition owing to increasing CCRT-induced toxicity, remains a clinical challenge for both patients and physicians, as these conditions delay treatment and lead to a poor quality of life at best, and to high healthcare costs and unexpected mortality at worst [3,4,5,6,7,8,9,10,11,12]. 

The mechanisms underlying the progression of malnutrition in patients with LAHNSCC can be attributed to inadequate energy intake and aberrant metabolism caused by varied degrees of systemic inflammation induced by cancer, treatment, or both [6,13,14]. The provision of sufficient energy intake and restriction of systemic inflammation are essential and reasonable approaches to prevent the development of malnutrition [4]. According to the European Society for Clinical Nutrition and Metabolism (ESPEN), the energy requirement for each patient with cancer is a total intake of at least 25–30 kcal/kg/day over the treatment course [15]. Nonetheless, this recommendation is not supported by sufficient evidence [15], and the effect of the recommended calorie intake during CCRT on the prognostic outcomes is seldom addressed. Additionally, the levels of nutrition–inflammation biomarkers (NIBs) can reflect the severity of systemic inflammation and can be used to clinically assess the malnourished status of patients with head and neck cancer at the pretreatment stage. The NIBs include nutrition indices (body weight, BW; body mass index, BMI; hemoglobin, Hb; albumin; total lymphocyte count, TLC; prognostic nutrition index, PNI; Patient-Generated Subjective Global Assessment, PG-SGA), and inflammation markers (C-reactive protein, CRP; neutrophil-to-lymphocyte ratio, NLR; platelet-to-lymphocyte ratio, PLR) [16,17,18,19,20,21,22]. Increasing evidence shows that the pretreatment levels of NIBs are correlated with prognosis and treatment outcomes in patients with head and neck cancer [16,17,18,19,20,21,22]. Because most of these studies were conducted retrospectively among heterogeneous patient populations with different tumor stages, tumor locations, treatment modalities, and sequences, the interpretation and clinical application of the results should be performed with caution. The correlations among the treatment interval changes in these NIBs, recommended daily calorie intake during the CCRT course, and prognostic outcomes remain elusive. Finally, along with decreasing calorie intake and increasing inflammation through the CCRT course, patients with LAHNSCC developed severe malnutrition, reflected by significant BW loss that is more specifically a change in the total body composition (TBC) [23,24,25,26,27]. Since changes in TBC are commonly observed among patients with cancer in response to aging, comorbidities, metabolic perturbations, and therapeutic intensity, they can be regarded as nutritional and inflammatory changes caused by the patient’s inherent characteristics, cancer, or treatment. Dual-energy X-ray absorptiometry (DXA) can be used to appropriately evaluate the status of three major components of TBC: lean body mass (LBM), total fat mass (TFM), and total body bone mineral content at the time of cancer diagnosis and during treatment [7,18,28,29,30]. In accordance with findings from other reports [7,25,31,32,33,34], we observed that the LBM and TFM assessed using DXA were significantly decreased among patients with LAHNSCC during CCRT [35,36]. Furthermore, a low pretreatment muscle mass is associated with increased treatment-related toxicity, early treatment failure, and recurrence-free survival among patients with LAHNSCC [35,36,37]. However, it remains unclear whether NIBs or DXA-derived parameters could affect the prognostic outcomes of patients provided the recommended daily calorie intake during the CCRT course. 

To address these concerns, we prospectively enrolled a homogenous group of patients with LAHNSCC (stage III, IVA, or IVB) and followed the ESPEN guidelines to provide each patient with an energy intake of at least 25 kcal/kg/day during the standard CCRT course. We also stratified the participants based on OCC with adjuvant CCRT and NOCC with primary CCRT for final analysis to offset the bias from different patient characteristics, treatment modality sequences, and different intent CCRTs with varied irradiation fields. Under the provision of the recommended daily calorie intake during the treatment, this study aimed to identify potential factors contributing to the 2-year mortality rate of patients with LAHNSCC receiving CCRT by simultaneously analyzing all covariates, including clinicopathological variables, blood NIBs, treatment-related profiles, and DXA-associated measurements. We assessed the 2-year mortality rate as the prognostic endpoint because no study has discussed the effects of the interplay among energy supply during CCRT, nutrition–inflammation status, and body composition on this outcome.

## 2. Materials and Methods

The ethics committee board of the Chang Gung Memorial Hospital, Taiwan, approved this study (approval numbers: 103-3365A3 and 201700158B0). We performed the study in accordance with the Good Clinical Practice Guidelines and the Declaration of Helsinki. Written informed consent was obtained from every participant before the study.

### 2.1. Patient Recruitment

This prospective study was conducted between February 2015 and July 2019 and enrolled eligible patients with histologically confirmed LAHNSCC originating from the oral cavity, oropharynx, hypopharynx, and larynx. 

LAHNSCC included stages III (T1-2, N1 or T3, N0-1), IVA (T4a, N0-1 or T1-4a, N2), and IVB (any T, N3 or T4b, any N), confirmed by the head and neck cancer committee of our institute according to the stage definition of the 7th edition of the American Joint Committee on Cancer staging system. Patients were aged less than 75 years, had an Eastern Cooperative Oncology Group (ECOG) performance status score of ≤2, showed adequate hematopoietic or organ function, and tested negative for human papilloma virus confirmed by p16-negative expression in tumor specimens. Patients were excluded if they had one of the following disorders: end-stage renal disease, decompensated liver cirrhosis with intractable ascites or hepatic encephalopathy, heart failure based on New York Heart Association Classification IV, autoimmune diseases, major gastrointestinal disorders, uncontrolled diabetes mellitus, or enduring infections. Patients receiving regular medications that could markedly interfere with their metabolism or weight, such as steroids or megestrol acetate, were also excluded. 

### 2.2. CCRT Schedule

Patients with OCC who met one of the following criteria received postoperative adjuvant CCRT: (1) positive surgical margin, (2) extranodal extension, or (3) at least three of the following risk factors: pT4, pN1, close margin ≤ 4 mm, invasion to blood vessel, lymphatic drainage, or perineural space, and poor histologic differentiation. Patients with unresectable NOCC disease for organ preservation underwent curative-intent primary CCRT. During the treatment course, we delivered radiotherapy (RT) at a dose of 64–72 Gy in 32 to 36 fractions over a 6- to 8-week period and concurrently administered chemotherapy with cisplatin (40 mg/m^2^) weekly. 

### 2.3. Provision of Recommended Daily Calorie during CCRT

Based on the recommendation of the ESPEN guidelines, we provided at least 25 kcal/kg/day, with percentage energy from carbohydrates/lipids in a 60:40 ratio, along with 1.0 g of protein/kg/day, to each patient during the CCRT course [15]. All patients received weekly dietitian visits for nutritional counselling and dietary record review. Feeding tube placement was mandatory if the BW loss was >5% or the daily calorie intake was less than 25 kcal/kg/day for 3 consecutive days during the CCRT course. The patients were provided oral nutritional supplements when they could not obtain the required daily calories from food. 

### 2.4. Clinicopathological Data and Blood NIBs

We collected clinicopathological data, including data on age, sex, body height, BW, ECOG performance status, comorbidities, tumor location, tumor stage, treatment protocol, and toxicity profiles, and records of substance exposure, including cigarette smoking, alcohol, and betel nut. BMI was calculated as the weight (in kilograms) divided by the height squared (in square meters) (expressed in kg/m^2^). The severity of comorbidities was assessed using the Head and Neck Charlson Comorbidity Index (HN-CCI) [38]. Smokers were considered if they were current cigarette smokers or had been previously exposed to cigarette smoke. Alcohol drinkers were considered if they consumed alcohol more than four times per week. Betel nut users were considered if they had used the substance in the preceding year. During the CCRT course, the RT dose was defined as the total radiation dose administered during CCRT, the RT duration was the number of days that patients needed to complete RT, and the cisplatin dose was the cumulative dose of cisplatin administered.

Blood NIBs, including hemoglobin (Hb, g/dL), white blood cell count (WBC, 10^3^/mm^3^), platelet count (10^3^/mm^3^), TLC (10^3^/mm^3^), albumin (g/dL), and CRP (mg/dL), were collected within 1 week before CCRT. TLC was calculated in terms of WBC (/mm^3^) × the percentage of lymphocytes in the blood. NLR was calculated as the ratio of the absolute neutrophil count to the lymphocyte count, and PLR was calculated as the ratio of the platelet count to the lymphocyte count. PNI = 10 × serum albumin (g/dL) + 0.005 × TLC (/mm^3^) [39]. 

The malnutrition status was also assessed based on BMI < 18.5 kg/m^2^, albumin < 3.5 g/dL, TLC < 1.5 × 10^3^ cells/mm^3^, or the PG-SGA. The scores ranged between 0 and 35, with scores 0–3 indicating stage A (well nourished), 4–8 indicating stage B (moderately malnourished), and ≥9 indicating stage C (severely malnourished) [40,41,42].

### 2.5. Assessment of Body Composition Parameters

TBC was evaluated by dual-energy fan-beam X-ray absorptiometry (Lunar iDXA, GE Medical System, Madison, WI, USA). The scanner software followed the body size and BMI and selected the scan mode. We used the enCORE Software, version 15 (GE Lunar, Chicago, IL, USA), to analyze the scans. We followed the guidelines set by the International Society for Clinical Densitometry to accurately position each participant [43]. The following parameters were obtained and analyzed: LBM, TFM, appendicular skeletal mass (ASM, arm and leg), and the distributions of android and gynoid fat. DXA-derived parameters were obtained 1 week before CCRT commencement and within 1 week of CCRT completion. 

Δ indicates the interval changes in the abovementioned blood NIBs and DXA-derived measurements before and after the treatment course.

### 2.6. Statistical Analysis

SPSS (version 22.0; SPSS Inc., Chicago, IL, USA) was used for statistical analyses. Based on a power of 80%, an α error of 0.05, and the head and neck cancer incidence rate in Taiwan, the calculated minimum sample size was 125. Considering the incomplete rate owing to intolerance to CCRT toxicity, reluctant follow-up to medical advice, poor compliance with data collection, or insufficient support from relatives, the estimated attrition rate was 30%, and the total number of patients would be increased to 169. Independent *t*-tests or Mann–Whitney tests were used for continuous variables, which were examined for normality before analysis. Chi-square tests were used for categorical variables. The 2-year mortality rate was defined as the proportion of patients who died within 730 days of the day of treatment initiation, which was used as the reference date owing to variations in the time for stage work-ups.

Because 14 parameters including BW, BMI, and DXA-derived parameters were highly correlated with each other, the variable number was reduced to minimize the loss of information. We utilized the principal axis factor with a varimax (orthogonal) rotation to assess four variables from BW and BMI, and ten parameters measured from DXA. The Kaiser–Meyer–Olkin (KMO) measure of sampling adequacy with a minimum acceptable value of KMO was 0.6, and factors with eigenvalues ≥1 were only considered. The variance percentage and factor score coefficient matrix were also calculated and presented. 

We used Cytoscape, an open-source software platform for creating two-dimensional visualizations, to decipher the relationships between the mean daily calorie intake during the CCRT course and the treatment interval changes in the blood NIBs and DXA-derived measurements. The attribute circle layout algorithm was weighted by the statistical significance of the correlations between the individual variables.

We applied Cox proportional hazards models with a forward stepwise selection to analyze the associations between different clinicopathological characteristics, treatment-related variables, toxicity profiles, NIBs, and DXA component parameters and the mortality rates in the univariate and multivariate analyses for different variables. All independent variables significantly associated with mortality rates (*p* ≤ 0.05) in the univariate analysis were included in the multivariate analysis. Variance inflation factors were used to test for collinearity. 

Receiver operating characteristic (ROC) curves were used to determine the optimal cutoff value when continuous variables showed significance in multivariate analysis. All differences in the mortality rates were considered statistically significant (two-tailed *p*-value < 0.05).

## 3. Results

### 3.1. Patient Characteristics

A total of 170 patients were recruited for this study, of which 123 patients were found to be eligible for the analysis at the end of the study. Sixty-nine patients belonged to the OCC subgroup, and fifty-four to the NOCC subgroup. The enrollment, allocation, treatment modality, and data collection schedule are shown in Figure 1. 

In both the OCC and NOCC subgroups, the patients were predominantly men with a median age of 53 to 55 years, and the majority of tumors were non-metastatic TNM stage IV. Most patients showed high proportions of substance exposure (cigarette smoking, alcohol, and betel nut) and at least one comorbidity. There was no statistical difference in the pretreatment levels and treatment interval changes in blood NIBs and DXA-derived parameters between the two subgroups (Table 1). Although the malnutrition rates varied according to different malnutrition assessment criteria, the OCC subgroup exhibited no statistically significant difference from the NOCC subgroup in the different criteria: PG-SGA-defined malnourished status (81.2% for OCC vs. 90.7% for NOCC, *p* = 0.328), BMI < 18.5 kg/m^2^ (20.3% for OCC vs. 20.4% for NOCC, *p* = 0.991), albumin < 3.5 g/dL (18.8% for OCC vs. 20.4% for NOCC *p* = 0.832), TLC < 1.5 × 10^3^ cells/mm^3^ (46.4% for OCC vs. 33.3% for NOCC, *p* = 0.144). 

The tongue was the most common tumor site (40.6%) in the OCC subgroup, whereas the hypopharynx was the most common tumor site (44.4%) in the NOCC subgroup. The OCC subgroup had a more advanced tumor size (T3 + T4: 88.4% for OCC vs. 63.0% for NOCC, *p* = 0.001), less regional lymph invasion (N2 + N3: 56.5% for OCC vs. 79.6% for NOCC, *p* = 0.007), and higher percentages of a poorly differentiated histological grade, betel nut use, and tracheostomy. During the CCRT course, compared to patients from the OCC group, patients from the NOCC subgroup required fewer feeding tube placements, fewer tube feeding days, lower calorie delivery, and a higher radiation intensity with a lower cisplatin dose; there was no difference in the protein intake between the two subgroups (Table 1). For treatment toxicities of grade 3 or higher, mucositis in the OCC subgroup and infection in the NOCC subgroup were the most common non-hematologic adverse effects. Neutropenia was the most commonly observed hematologic counterpart in both subgroups. With respect to the treatment outcome, the OCC group showed a trend of a higher 2-year mortality rate than the NOCC subgroup (43.5% vs. 29.6%, *p* = 0.115, Table 1).

### 3.2. Comparison of Clinicopathological Variables, NIBs, Treatment Factors, Body Composition, and Prognosis in Patients Enrolled in Two Different Daily Calorie Supply Programs during CCRT

To investigate whether a higher daily energy intake is beneficial to treatment outcomes, including toxicity and prognosis, we stratified patients into two different energy supply programs (25–30 and ≥30 kcal/kg/day), based on a cutoff of 30 kcal/kg/day, in accordance with the ESPEN recommendations [15] (Table 2). 

In both the OCC and NOCC subgroups, patients enrolled in the two energy supply programs showed no difference in the percentage of tube feeding placement, but patients receiving ≥30 kcal/kg/day had a significantly higher energy and protein supply during the CCRT course (OCC: 38.6 ± 7.0 vs. 25.2 ± 4.3 kcal/kg/day, *p* < 0.001, and 1.3 ± 0.4 vs. 0.9 ± 0.9 g/kg/day, *p* = 0.014; NOCC: 36.6 ± 7.8 vs. 25.7 ± 3.8 kcal/kg/day, *p* < 0.001, and 1.2 ± 0.8 vs. 0.8 ± 0.9 g/kg/day, *p* = 0.021). Further, we noted a longer feeding tube placement duration in patients receiving ≥30 kcal/kg/day than in patients receiving 25–30 kcal/kg/day in the OCC subgroup (28.2 ± 4.1 vs. 48.8 ± 5.7, *p* = 0.007), but not in the NOCC subgroup (20.6 ± 4.0 vs. 27.5 ± 9.5, *p* = 0.451) (Table 2). 

In both the OCC and NOCC subgroups, there was no difference among the clinicopathological variables, including age, sex, tumor stage, performance status, histologic grade, substance exposure, comorbidity, presence of tracheostomy, and PG-SGA, with diagnosis based on the two energy supply programs. In addition, none of the blood NIB data, including pretreatment and treatment interval changes, were different between the two energy supply programs, except for higher levels of treatment interval NLR changes in patients from the OCC group supplied with 25–30 kcal/kg/day. Furthermore, there was no difference in CCRT intensity, including the dose, fractions and dose of radiation, and accumulative dose of cisplatin used in the CCRT course, and grade 3/4 CCRT-associated toxicity between the two energy supply programs (Table 2). 

Although patients from both the OCC and NOCC subgroups receiving ≥30 kcal/kg/day showed lower BW, BMI, and DXA parameters than patients receiving 25–30 kcal/kg/day before CCRT, they developed fewer interval changes in these measurements (with the exception of ASM and gynoid fat percentage) during the CCRT course than patients receiving 25–30 kcal/kg/day. There was a higher 2-year mortality rate in patients receiving ≥30 kcal/kg/day than in patients receiving 25–30 kcal/kg/day (OCC: 57.1 vs. 37.1%, *p* = 0.045; NOCC: 54.5 vs. 23.3%, *p* = 0.043) (Table 2).

### 3.3. Factor Analysis

Ten DXA-derived parameters were stratified into two main classes: the body muscle class (LBM, ASM, ΔLBM%, and ΔASM%) and the body fat class (TFM, android, gynoid, ΔTFM%, Δandroid%, and Δgynoid%). Principal axis factor analysis with varimax rotation generated a four-factor solution. Table 3 shows the rotated component matrix, eigenvalues, and percentage of variance explained, with the KMO measure of sampling adequacy as 0.749. The four components explain 90.5% of the variance in the OCC subgroup and 93.8% of that in the NOCC subgroup.

In the OCC subgroup, three items loaded in Factor 1: all were related to total body fat (TBF) mass before CCRT (TFM, android, and gynoid) and explained 45.1% of the variance. Three items loaded in Factor 2: all were related to the treatment interval change in TBF (ΔTFM%, Δandroid%, and Δgynoid%) and explained 22.1% of the variance. Two items loaded in Factor 3: both were related to the treatment interval change in total body muscle (TBM) (ΔLBM% and ΔASM%) and explained 12.7% of the variance. Two items loaded in Factor 4: both were related to TBM storage before CCRT and explained 10.6% of the variance (Table 3).

In the NOCC subgroup, two variables (LBM and ASM) loaded in Factor 1: both were related to TBM storage before CCRT and accounted for 48.2% of the variance. Three variables (TFM, android, and gynoid) loaded in Factor 2: all were related to the TBF mass before CCRT and accounted for 20.4% of the variance. Two variables (ΔLBM% and ΔASM%) loaded in Factor 3: both were related to the interval change in TBM during the CCRT course and accounted for 14.2% of the variance. Three variables (ΔTFM%, Δandroid%, and Δgynoid%) loaded in Factor 4: all were related to the interval change in TBF during the CCRT course and accounted for 10.5% of the variance (Table 3).

### 3.4. Correlation between Mean Daily Calorie Supply during CCRT Course and Treatment Interval Changes in the Blood NIBs and DXA-Derived Factors

According to the factor analysis of DXA-derived measurements in the present study, there were two individual factors representing the treatment interval changes in the TBM and TBF for each subgroup: Factor 2 (fat change) and Factor 3 (muscle change) for OCC; Factor 3 (muscle change) and Factor 4 (fat change) for NOCC (Table 3).

For OCC, the mean daily calorie intake during the CCRT course correlated with treatment interval changes in both TBF and TBM mass (Factor 2 and Factor 3) but was not associated with the blood NIBs. The platelet count change correlated with the TBM changes (Factor 3). Treatment interval changes in certain blood NIBs were correlated with each other: Hb with PLR or PNI; WBC with platelet count or CRP; platelet with PLR; TLC with NLR, PLR, or PNI; albumin with PNI; NLR with PLR; PLR with PNI (Figure 2a). 

For NOCC, the mean daily calorie intake correlated with treatment interval changes in the TBM (Factor 3) and TBF (Factor 4) and the CRP levels. Total muscle change (Factor 3) was correlated with changes in the albumin and CRP levels. Similar to that observed in the OCC subgroup, treatment interval changes in certain blood NIBs were correlated with each other in the NOCC subgroup (changes in Hb were correlated with those in the WBC count, albumin level, or PNI; changes in the WBC count were correlated with those in platelets or CRP; changes in TLC were correlated with those in NLR, PLR, or PNI; changes in albumin were correlated with those in PNI; and changes in NLR were correlated with those in PLR (Figure 2b)).

### 3.5. Factors Independently Contributing to the 2-Year Mortality Rate of Patients with LAHNSCC Receiving CCRT

Of the 69 patients with OCC, 30 (43.5%) died within 730 days from the beginning of CCRT. Twenty-two patients died owing to tumor progression, and eight died owing to non-cancer etiology associated with sepsis from pneumonia. The mean age was 53.0 years (range, 30–67 years), the mean time to death was 14.5 months (range, 4.0–21.9 months), and the mean daily calorie supply during the CCRT course was 29.6 ± 1.6 kcal/kg/day. After adjusting for all covariates, including clinicopathological variables, mean daily calorie supply during CCRT, treatment-related factors, blood NIBs, and body composition parameters, we found that three variables, namely, performance status, PLR, and Factor 3 (treatment interval TBM change), independently contributed to the 2-year mortality rate in the multivariate analysis (Table 4). Patients with a suboptimal performance status, high PLR, and less total muscle loss (high OCC Factor 3) during CCRT had a higher 2-year mortality rate (Figure 3).

Of the 54 patients with NOCC, 16 (29.6%) died within 730 days from the beginning of CCRT, all of whom died owing to tumor progression. The mean age was 59.7 years (range, 49–74 years). The mean time to death was 17.1 months (range, 6.1–21.5 months), and the mean daily calorie supply during the CCRT course was 26.8 ± 1.1 kcal/kg/day. Similar to that observed in patients with OCC, multivariate analysis showed that age, NLR, and Factor 2 (pretreatment TBF) independently affected the 2-year mortality rate of patients with NOCC (Table 4). Patients with an older age, a high NLR, and less body fat storage (low NOCC Factor 2) before CCRT showed a higher 2-year mortality rate (Figure 3). However, we failed to confirm the independent effect of the mean daily calorie supply during the CCRT course on the 2-year mortality rate of patients with OCC or NOCC receiving CCRT (Table 4).

## 4. Discussion

The provision of the recommended daily calorie intake during CCRT is the most pressing issue related to nutrition for patients with LAHNSCC who need CCRT as postoperative adjuvant therapy (patients with OCC) or primary therapy (patients with NOCC). All efforts of clinical treatment, including nutritional counselling, oral nutritional supplements, feeding tube placement, and parenteral nutrition, are intended to achieve a mandatory calorie supply to prevent malnutrition [44]. Nonetheless, it remains unknown if patients gain clinical benefits, such as lower treatment-associated toxicity and better survival outcomes, with the recommended calorie intake during CCRT. Since LAHNSCC is a heterogeneous disease with different patient characteristics and therapeutic intents, and the differential effects of calorie intake on nutrition-related issues and prognosis require objective assessment, the current study stratified patients into two subgroups: OCC with postoperative adjuvant CCRT, and NOCC with primary curative-intent CCRT (Table 1). In our analysis, the OCC subgroup had a higher percentage of patients with feeding tube placement, a longer tube feeding duration, and greater calorie delivery than the NOCC subgroup during the CCRT course (Table 1). Patients with OCC or NOCC showed no difference in clinicopathological variables, pretreatment levels of and treatment interval changes in NIBs, percentage of feeding tube placement, CCRT intensity, and treatment-associated toxicities based on the 25–30 and ≥30 kcal/kg/day energy supply programs (Table 2). In both the OCC and NOCC subgroups, the mean daily calorie supply correlated with the interval changes in TBM and TBF during CCRT, but not with the treatment interval changes in NIBs (Figure 2). However, patients receiving ≥30 kcal/kg/day had lower pretreatment levels of BW, BMI, and DXA parameters but showed greater protein intake and developed fewer interval changes in the anthropometric index and DXA measurements over the CCRT course than patients receiving 25–30 kcal/kg/day during CCRT (Table 2). Although the univariate analysis showed a significant effect of the ≥30 kcal/kg/day program on the 2-year mortality rate, multivariate analysis failed to confirm this impact. The pretreatment inflammation status and body composition parameters assessed using DXA independently contributed to the 2-year mortality rate of patients with LAHNSCC (Table 4). To our knowledge, this prospective study is the first to demonstrate that a greater energy intake in patients with LAHNSCC during CCRT led to lesser reductions in BW, BMI, and TBC during the CCRT course but was not associated with treatment interval changes in NIBs or with prognostic outcomes, such as the 2-year mortality rate. 

The energy requirement for patients with LAHNSCC undergoing CCRT has been debated for the past three decades [7,45,46,47,48,49,50]. Van der Berg et al. used the Food Frequency Questionnaire and found that the energy intake significantly decreased from 33.3 ± 13 kcal/kg/day at the beginning to 19 ± 8 kcal/kg/day at the end of CCRT, corresponding to an estimated reduction of 122 kcal/day and, consequently, an energy deficit of more than 50,000 calories during the treatment course [50]. Kubrak et al. applied a 3-day dietary record and reported that, compared to that at the beginning of CCRT, patients had a lower energy intake (13.2 kcal/kg/day) during the CCRT course [47]. Using 24 h dietary recall and indirect calorimetry, de Carvalho et al. showed that throughout the CCRT course, patients had a lower calorie intake (200–300 kcal/day), and the resting metabolic rate was reduced by 2.67 kcal/kg/24 h under BW adjustment [45]. However, using indirect calorimetry and data from dietary records, Langius and Ng et al. found that after adjusting for BW or fat-free mass, there was no difference in energy intake or BW-adjusted resting energy expenditure (REE) before and after CCRT [48,49]. Garcia-Peris et al. further showed that the REEs of 18 patients with LAHNSCC followed a U-shaped curve, with the rates being the highest at the beginning and the end of treatment and 2 weeks after the treatment [46]. The discrepancies among the study findings could be attributed to the small sample size, ethnic differences, recall bias in dietary intake, or investigation in mixed patient populations with varied stages and different treatment modalities. Particularly in assessments of nutrition or energy intake, either personal recall errors for quantitative dietary record review or the lack of a uniform food questionnaire for qualitative assessments may limit the data accuracy and their application in daily practice. On the contrary, REE may be the gold standard for the measurement of energy needs during CCRT because it represents the largest component accounting for nearly 70% of the total daily energy expenditure and is objectively measured by indirect calorimetry [15]. The measured REE of patients with head and neck cancer during the CCRT course is approximately 22 kcal/kg/day [45,46,49]. Therefore, the estimated daily energy requirement for these patients may be approximately 31 kcal/day or higher, which is compatible with the recommendations provided in the guidelines of the United Kingdom and academic nutrition societies [15,51] and reports published by experienced researchers [52,53,54], but is relatively higher than the consensus criteria from the American Society for Parenteral and Enteral Nutrition (ASPEN) (≥25 kcal/kg according to actual BW) [55]. 

With respect to the quantity of protein administered during the CCRT course, patients with head and neck cancer showed a lower protein intake (50 g/day) during the treatment course compared to that at the beginning of CCRT [47]. Patients from the OCC or NOCC subgroup consumed an average of 1.0 g of protein/kg/day during the CCRT course, but patients enrolled in the ≥30 kcal/kg/day energy supply program consumed 1.2 g of protein/kg/day, which met the recommended protein intake (1.2–2.0 g/kg/day under ambulatory cancer patients based on both ASPEN and ESPEN guidelines [15,55]). Therefore, an increased protein intake may correspond to a higher calorie supply during the CCRT course. Additionally, some reports showed that supplements with certain amino acids might improve protein synthesis in cancer with cachexia [15,56], but the beneficial effect of this modulation in protein quality on daily protein intake remains unknown in patients with head and neck cancer receiving CCRT. 

Some reports have shown the positive effects of an energy intake of 30 to 35 kcal/kg/day and a protein intake of 1.2 to 2.0 g/kg/day in patients with head and neck cancer during CCRT on the maintenance of BW and improvement of serum albumin levels during treatment [32,52,54,57,58], but the effect on prognosis has seldom been assessed [54,57]. Patients who consumed ≥33 kcal/kg/day and ≥1.3 g protein/kg/day could maintain their BW during the CCRT course [32,52,58]. Meanwhile, Giles et al. pointed out that despite different energy and protein intakes, patients receiving ≥30 kcal/kg/day and ≥1.2 g of protein/kg/day showed a BW loss percentage similar to that of patients receiving <30 kcal/kg/day and <1.2 g protein/kg/day during the CCRT course [52]. Daly et al. analyzed 40 patients with locally advanced nasopharynx and oropharynx cancers receiving RT. Compared with patients consuming 30 kcal/kg/day, patients consuming 39 kcal/kg/day showed a greater protein intake and lesser BW loss during treatment; however, there was no survival difference between the two calorie intake subgroups [57]. A prospective population-based cohort study enrolled 1756 all-stage patients with head and neck cancer undergoing various therapeutic modalities. In the study, 40% of the patients were older than 65 years, and nearly 30% of the patients were female and had stage I-II disease. The authors found that patients consuming ≥34 kcal/kg/day were able to maintain BW and may have gained a survival advantage, whereas patients with a mean calorie intake of 30 kcal/kg/day failed to achieve either clinical benefit [54]. These inconsistent results could be ascribed to the study design with a heterogeneous patient population or the lack of comprehensive analysis owing to the lack of adjustment of covariates, such as treatment-related toxicity, anthropometric parameters, and body composition parameters. The current study enrolled a homogenous patient population with LAHNSCC undergoing an identical CCRT course; a total of 96.7% of the patients were male, and more than 85% of the patients were aged less than 65 years. We comprehensively assessed the effects of calorie intake on BW, body composition, NIBs during treatment, and the 2-year mortality rate. After adjustments were made for all possible confounding covariates, our results clearly showed that in the OCC and NOCC subgroups, compared to patients with a lower daily calorie intake (25–30 kcal/kg/day), patients with a higher daily calorie intake (≥30 kcal/kg/day) had fewer percentages of changes in BW, BMI, and body composition parameters but showed no advantage in 2-year mortality. These findings, inclusive of ours, suggest that a higher energy supply during CCRT in patients with LAHNSCC provided with the recommended calorie intake could offer an advantage in BW maintenance during treatment but might fail to prove its positive influence on the prognostic outcomes.

Although the energy support provided during the CCRT course may induce positive nutritional benefits, such as BW maintenance [59,60], it remains challenging to reverse the process of malnutrition [61] or improve metabolic aberrations [62] simply via energy intake, because malnutrition is multifactorial and may be more closely associated with inflammatory processes and a defect in orexigenic signals [63]. Additionally, symptoms of treatment-induced toxicity, such as pain, dysphagia, mucositis, or emesis, increase the daily energy expenditure and, subsequently, necessitate intense nutrition intake above the recommended energy intake [11,54]. To fulfill escalating energy needs, inappropriate feeding methods, such as bolus feeding in a short time interval, are often adopted in daily practice. However, these may counteract the beneficial effects of nutritional intake during treatment and may occasionally lead to unexpected aspiration pneumonia, diarrhea, and sepsis. Furthermore, the current study showed that patients consuming ≥30 kcal/kg/day had lower pretreatment values in BW, BMI, and DXA parameters but exhibited higher 2-year mortality rates than those receiving 25–30 kcal/kg/day, implying that excess energy supply beyond the recommended calorie intake during CCRT may be the preference of patients and healthcare professionals concerning a lower pretreatment nutrition status that had a negative impact on prognosis and might be deteriorated over the treatment course if not treated [24]. Lastly, our previous study showed that though more than 75% of patients with head and neck cancer cachexia were unable to meet the mandatory daily calorie requirement during the treatment period, patients could maintain their BW and improve serum albumin levels during palliative chemotherapy when administered immune-modulated oral nutritional formula containing omega-3 fish oil and selenium [64]. Therefore, the calorie supply during CCRT may be necessary to maintain BW but remains insufficient for improving treatment outcomes and prognosis in patients with advanced cancers. From our perspective, under appropriate calorie intake, the inflammation status resulting from a pre-existing tumor and/or ongoing treatment may play a key role in malnutrition development during CCRT and the prognosis of patients with LAHNSCC. The current study clearly demonstrates that two pretreatment inflammatory markers, PLR and NLR, independently contributed to and were positively correlated with the 2-year mortality rates of patients with OCC and NOCC, respectively. Inflammation biomarkers, such as CRP, NLR, and PLR, are considered to reflect the complicated systemic inflammatory response resulting from the interaction between host immunity and the tumor microenvironment and help determine the malnutrition status [65] and predict the prognosis of patients with head and neck cancer [16,18,22,66,67,68]. Two reports showed the independent prognostic effect of pretreatment PLR on the overall survival of patients with OCC undergoing surgery with RT or CCRT [67,69]. Acharya et al. further showed that preoperative PLR was associated with lymph node involvement and predicted lymph node metastasis in 68 patients with OCC undergoing surgery more effectively than NLR [66]. Similar to the prognostic role of PLR in OCC, a positive association between an elevated NLR and increased hazard ratios of overall survival in patients with NOCC was shown in two meta-analyses [22,68]. The pretreatment NLR showed an independent effect on progression-free and overall survival in patients with NOCC who underwent primary CCRT [70,71,72,73]. Therefore, our results support the notion that the pretreatment systemic inflammatory status represented by PLR and NLR could affect the prognostic outcomes of patients with head and neck cancer undergoing CCRT. 

The pretreatment systemic inflammatory status was also associated with the nutritional condition, BW, and body composition of patients with advanced gastrointestinal and lung cancer [65,74,75,76]. Several hormones and cytokines, such as leptin, ghrelin, interleukin (IL)-1β, IL-6, and tumor necrosis factor alpha, contribute to the inflammation-mediated loss of fat and muscle and actively participate in the homeostasis of muscle and fat tissues in patients with cancer [77,78]. These observations support the results of the present analysis, which showed that the LBM and TFM of patients with LAHNSCC before CCRT were correlated with several pretreatment NIBs (Appendix A). Taken together, the pretreatment systemic inflammatory status affects the body composition before CCRT as well as the 2-year mortality of patients with LAHNSCC. Conversely, the correlation between interval changes in NIBs and body composition changes during the treatment course and their interactive effect on prognosis in patients with cancer have not been investigated thoroughly. Furthermore, Baxi et al. proposed that body composition change is common during anticancer treatment but is not generally prognostic because patients may have insufficient calorie intake [24]. To address these uncertainties, in the current study, we adopted calorie supply programs recommended by academic societies for patients with LAHNSCC undergoing CCRT and found that the change in TBM during CCRT correlated with that in certain NIBs (OCC Factor 3 correlated with Δplatelet count, and NOCC Factor 3 correlated with Δalbumin and ΔCRP). In addition, the mean daily calorie intake during CCRT was associated with changes in TBM and TBF, which is compatible with our previous observation [35]. Finally, we showed that the treatment interval change in TBM (OCC Factor 3) independently contributed to the 2-year mortality rate. Hence, the treatment interval changes in NIBs could be associated with changes in body composition parameters, which were also correlated with energy intake during the treatment course and possibly affected the prognostic outcome, such as the 2-year mortality rate, of these patients.

Another intriguing phenomenon was noted in the correlation between the treatment interval change in TBM (OCC Factor 3) and the 2-year mortality rate of patients with OCC who received adjuvant CCRT. Patients with a higher OCC Factor 3 showed less treatment interval TBM loss, including the loss of both LBM and ASM, but they also showed more deaths within 2 years (Appendix A and Figure 3). This observation is different from that of previous studies reporting that pretreatment or treatment interval muscle loss induced by cancer or chemotherapy is associated with poor treatment outcomes in patients with cancer [79,80,81,82,83]. The patients were older, received a greater daily calorie supply, underwent feeding tube treatment for a longer duration, and even failed to develop TFM loss during CCRT, as compared to patients with a low OCC Factor 3 (Appendix A). Although the effect of treatment interval muscle loss on survival outcome could vary in different cancer types and treatment regimens, this discrepancy indicates that a stereotypically negative association between muscle loss and prognosis overlooks the fact that the maintenance of optimal functioning and vitality of muscle mass requires efficient protein turnover when the body is constantly challenged by mechanical, chemical, and oxidative stresses [84]. From our perspective, the current study shows that under the recommended energy supply during the treatment course, muscle breakdown during CCRT in patients with a low OCC Factor 3 may offer an alternative method for maintaining optimal muscle function; moreover, through the muscle breakdown process, excess energy could be generated in these patients to cope with the energy deficit from increasing oxidative stress induced by CCRT, since muscle tissues are the primary energy reserve [85]. We further speculate that despite a greater calorie supply and longer duration of feeding tube placement during the CCRT course, patients with a high OCC Factor 3 may lose the ability to compensate for treatment-induced energy deficit by acquiring additional energy support from muscle or fat breakdown or may fail to efficiently convert ingested food to energy, possibly owing to an aberrant metabolism caused by cancer [86,87,88] or cancer treatment [89,90,91].

In the case of NOCC, as compared with patients with a low NOCC Factor 2, patients with a high NOCC Factor 2 showed greater pretreatment TBF storage, including TFM, android and gynoid distributions, and higher values of BMI, BW, Hb, TLC, albumin, PNI, LBM, and ASM at the beginning of CCRT (Appendix A). Therefore, the patients had a clinically better nutritional status and preferentially selected oral intake instead of tube feeding. Although the patients consumed fewer calories during the CCRT course, and more than two-thirds of the patients (14 out of 20) did not receive feeding tube placement, the patients showed a lower 2-year mortality rate (Appendix A and Figure 3). This finding supports our viewpoint that increased energy supply alone during the CCRT course may not always improve the prognostic outcome. Furthermore, these patients experienced significant reductions in the Hb levels and TFM loss during treatment. This phenomenon could be partially explained by the following speculation. On one hand, patients with a high NOCC Factor 2 had a more severe inflammation status, indicated by a higher pretreatment PLR level at the beginning of CCRT (Appendix A), which may have aggravated anemia and induced greater lipolysis [78,92]; on the other hand, these patients may have obtained extra energy from the elevated lipolytic process to compensate for the energy shortage caused by CCRT and the lower calorie intake over the treatment course. Lastly, Pai et al. retrospectively evaluated the effect of body composition, assessed using computed tomography imaging, on the survival outcomes of 881 patients with LAHNSCC, 95% of whom belonged to the NOCC group. They found that patients with a higher subcutaneous adipose tissue index receiving curative-intent CCRT had superior locoregional control and overall survival, whereas the skeletal muscle index did not predict these outcomes [93]. Hence, these findings further reinforce our observation that pretreatment TBF storage (NOCC Factor 2), and not TBM, affects the 2-year mortality rate of patients with NOCC undergoing primary CCRT.

Performance status and age are well-known predictors of survival outcomes in patients with head and neck cancer undergoing RT or chemoradiotherapy [17,19,70,93,94]. Similarly, performance status and age were independent predictors of the 2-year mortality rate of patients with OCC and NOCC, respectively, in our analysis. For OCC with adjuvant CCRT, compared to patients with a good performance status (ECOG < 2), patients with a suboptimal performance status (ECOG = 2) had a high percentage of tracheostomy, a longer duration of feeding tube placement, an inferior nutritional status (low levels of pretreatment BMI, BW, albumin, LBM, and ASM), a more severe inflammatory status (high levels of pretreatment NLR and PLR), a lower radiation dose, and greater levels of grade 3/4 pharyngitis (Appendix A). For NOCC with primary CCRT, compared to younger patients (aged ≤ 52.5 years), older patients (aged > 52.5 years) had a weaker nutritional status (low values of pretreatment BW, PNI, LBM, and ASM) and a more severe inflammatory status (high level of pretreatment NLR) and received a lower cisplatin dose (Appendix A). These specific clinical features may explain why patients with risk factors, such as a suboptimal performance status and older age, may have a higher 2-year mortality rate. 

This study has several limitations. First, the enrolled participants were Taiwanese, and most were men (96.7%). The results with respect to the ethnic groups, female gender, and regional vulnerability to this disease remain uncertain. Hence, the current findings should be carefully applied to non-Taiwanese patients, different treatment schedules, and nutrition support plans. Second, the current study discussed the effects of calorie supply programs on clinical outcomes with no appropriate adjustment for protein intake, including the quantity and quality of protein. Participants of the current study had an average protein intake of 1.0 g/kg/day during the CCRT course, which was obviously lower than the recommended protein intake (1.2–2.0 g/kg/day) [15,55]. It remains to be investigated whether increased protein intake or modulation of protein quality could affect treatment outcomes. Finally, it should be noted that data from 42 patients (25.4%) could not be analyzed owing to an incomplete CCRT course or incomplete data collection during treatment. The aim of this study was to assess the treatment interval TBC change in patients with LAHNSCC who were able to complete CCRT; therefore, patients who failed to complete the course and comply with the data collection schedule were not considered eligible for the final analysis. In the OCC subgroup, one patient was medically unfit owing to poliomyelitis, ten patients discontinued the treatment course owing to unexpected severe sepsis (four patients) and drop-out (six patients), and eight patients failed to complete data collection owing to delayed scheduled DXA measurements or missing blood tests. In the NOCC subgroup, 15 patients with an incomplete CCRT course developed ischemic heart disease (2 patients), acute renal failure (1 patient), and severe pneumonia (6 patients) or were unwilling to continue CCRT treatment (6 patients), and data collection was incomplete for 8 patients, because 6 patients missed the scheduled DXA examination and 2 patients were lost to follow-up after CCRT completion. Although a small number of patients failed to meet the criteria for final analysis, there was no statistical difference in baseline characteristics between the complete CCRT and incomplete CCRT data collection groups, except for the PG-SGA status (Appendix A). From our perspective, the impact of the incomplete subgroup on the current analysis may be trivial, if present.

## 5. Conclusions

This prospective observational study demonstrates that under the recommended energy supply (≥25 kcal/kg/day) during the treatment course, a greater energy supply led to lower muscle and fat mass loss; however, it did not reduce the occurrence of treatment-associated toxicity, nor did it affect the 2-year mortality rate of patients with LAHNSCC receiving CCRT. Rather, in addition to the patient characteristics (performance status for OCC and age for NOCC), the inflammation status and body composition contributed to the 2-year mortality rate.

## Figures and Tables

**Figure 1 biomedicines-10-00388-f001:**
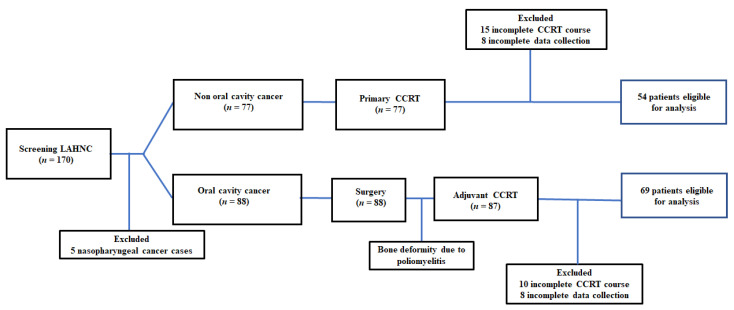
Flow diagram of recruitment. Patients who dropped out during the treatment or failed to receive at least four cycles of weekly cisplatin (40 mg/m^2^) concurrently with planned radiotherapy (64–72 Gy) were considered as the incomplete CCRT subgroup. Patients who did not complete required DXA examinations or missed scheduled blood tests were considered as the incomplete data subgroup. LAHNC, locally advanced head and neck cancer; DXA, dual-energy X-ray absorptiometry; CCRT, concurrent chemoradiotherapy.

**Figure 2 biomedicines-10-00388-f002:**
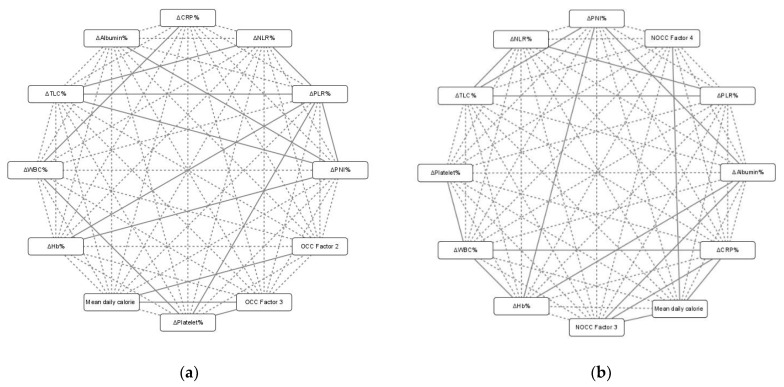
(**a**) Patients with OCC with postoperative adjuvant CCRT; (**b**) patients with NOCC with primary CCRT. Two-dimensional visualization of the correlations between mean daily calorie intake (at least 25 kcal/kg/day) during the CCRT course, treatment interval changes in nutrition–inflammation biomarkers (NIBs) (ΔHb%, ΔWBC%, ΔPlatelet%, ΔAlbumin%, ΔTLC%, ΔCRP%, ΔNLR%, ΔPLR%, and ΔPNI%), and body composition (OCC Factor 2, OCC Factor 3, NOCC Factor 3, and NOCC Factor 4). OCC, oral cavity cancer; NOCC, non-oral cavity cancer; Hb, hemoglobin; WBC, white blood cell count; TLC, total lymphocyte count; CRP, C-reactive protein; NLR, neutrophil-to-lymphocyte ratio; PLR, platelet-to-lymphocyte ratio; PNI, prognostic nutritional index. Δ indicates the interval changes in the abovementioned blood NIBs before and after the CCRT course. Individual variables are represented by the nodes, and their associations are represented by the edges (connecting lines). Solid edges indicate significant associations between two variables (*p* < 0.05). Dashed edges indicate no correlation between two variables.

**Figure 3 biomedicines-10-00388-f003:**
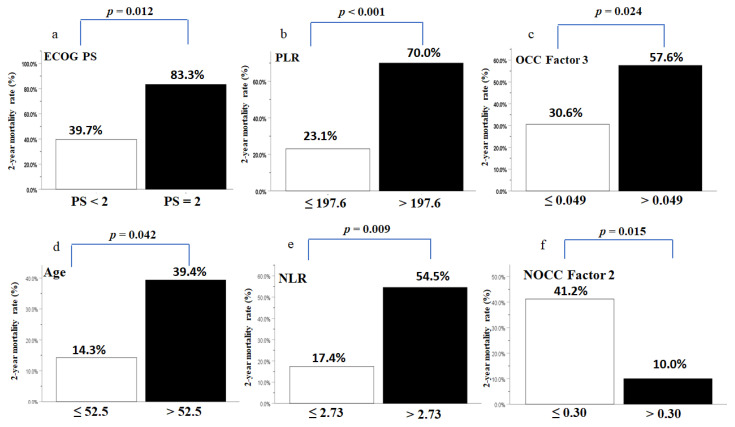
(**a**–**c**) Two-year mortality rates of patients with OCC with postoperative adjuvant CCRT stratified by ECOG PS (performance status), PLR, and OCC Factor 3 (change in body muscle mass during CCRT); (**d**–**f**) two-year mortality rates of patients with NOCC with primary CCRT stratified by age, NLR, and NOCC Factor 2 (body fat mass before CCRT). OCC, oral cavity cancer; NOCC, non-oral cavity cancer; NLR, neutrophil-to-lymphocyte ratio; PLR, platelet-to-lymphocyte ratio. The cutoff value analyzed by ROC curves for PLR was 197.6 (area under the curve (AUC): 0.807, *p* < 0.001), that of OCC Factor 3 was 0.049 (AUC: 0.692, *p* = 0.01), that of age was 52.5 (AUC: 0.708, *p* = 0.017), that of NLR was 2.73 (AUC: 0.689, *p* = 0.037), and that of NOCC Factor 2 was 0.30 (AUC: 0.676, *p* = 0.046).

**Table 1 biomedicines-10-00388-t001:** Baseline and treatment characteristics of 69 patients with OCC treated with postoperative adjuvant CCRT and 54 patients with NOCC treated with primary CCRT.

	OCC with Postoperative Adjuvant CCRT	NOCC with Primary CCRT	
Variables, Expressed as Numbers (%), Percentage (%), or Mean ± SD (Median)			*p*-Value *
**Patient number**	69 (56.1)	54 (43.9)	
**Clinicopathologic variables**			
**Sex**			0.203
**Male**	68 (98.6)	51 (94.4)	
**Female**	1 (1.4)	3 (5.6)	
**Age (years) (median)**	53.1 ± 8.4 (53.0)	55.3 ± 8.5 (54.7)	0.184
**<65**	62 (89.9)	46 (85.2)	0.432
**≥65**	7 (10.1)	8 (14.8)	
**Tumor subsites (OCC/NOCC)**			
**Buccal mucosa/Tonsil**	20 (29.0)	13 (24.1)	
**Tongue/Tongue base**	28 (40.6)	6 (11.1)	
**Gingiva/Soft palate**	13 (18.9)	3 (5.6)	
**Mouth floor/Hypopharynx**	3 (4.3)	24 (44.4)	
**Retromolar/Larynx**	2 (2.9)	8 (14.8)	
**Lip**	2 (2.9)	---	
**Hard palate**	1 (1.4)	---	
**TNM Stage**			0.613
**III**	4 (5.8)	5 (9.3)	
**IVA**	50 (72.5)	35 (64.8)	
**IVB**	15 (21.7)	14 (25.9)	
**T status (%)**			0.001 *
**T0-2**	11.6	37.0	
**T3-4**	88.4	63.0	
**N status (%)**			0.007 *
**N0-1**	43.5	20.4	
**N2-3**	56.5	79.6	
**Histological grade (%)**			0.023 *
**Well differentiated**	11.6	3.7	
**Moderately differentiated**	73.9	63.0	
**Poorly differentiated**	14.5	33.3	
**Smoking user (%)**	91.3	90.7	0.913
**Alcohol user (%)**	73.9	75.9	0.799
**Betel nut user (%)**	76.8	50.0	0.002 *
**HN-CCI (%)**			0.760
**0**	42.1	40.7	
**≥1**	57.9	59.3	
**ECOG performance status (%)**			0.046 *
**0**	2.9	11.1	
**1**	86.4	87.0	
**2**	8.6	1.9	
**Tracheostomy (%)**	66.7	18.5	<0.001 *
**PG-SGA assessment before CCRT (%)**			0.328
**Stage A (well nourished)**	18.8	9.3	
**Stage B (moderately malnourished)**	55.1	61.1	
**Stage C (severely malnourished)**	26.1	29.6	
**Anthropometric data and blood NIB data**			
**Before CCRT**			
**BW (kg) (median)**	63.6 ± 12.6 (61.1)	62.1 ± 12.1 (61.9)	0.529
**BMI (kg/m^2^) (median)**	22.7 ± 4.3 (21.6)	22.7 ± 4.0 (22.5)	0.966
**<18.5 (%)**	20.3	20.4	0.991
**≥18.5 (%)**	79.7	79.6	
**Hb (g/dL) (median)**	11.7 ± 1.5 (11.5)	11.9 ± 1.7 (11.7)	0.459
**WBC (×10^3^ cells/mm^3^) (median)**	7.3 ± 2.5 (7.3)	7.1 ± 2.9 (6.9)	0.731
**Platelet count (×10^3^/mm^3^) (median)**	341.1 ± 148.4 (327)	251.4 ± 75.3 (241)	<0.001 *
**TLC (×10^3^ cells/mm^3^) (median)**	1.6 ± 0.6 (1.4)	1.8 ± 0.7 (1.7)	0.205
**<1.5 (%)**	46.4	33.3	0.144
**≥1.5 (%)**	53.6	69.7	
**Albumin (g/dL) (median)**	3.8 ± 0.6 (3.8)	3.8 ± 0.5 (3.7)	0.720
**<3.5 (%)**	18.8	20.4	0.832
**≥3.5 (%)**	81.2	79.6	
**CRP (mg/dL) (median)**	11.2 ± 1.8 (7.7)	18.7 ± 6.6 (5.9)	0.223
**NLR (median)**	3.7 ± 4.2 (3.0)	2.9 ± 2.9 (2.3)	0.241
**PLR (median)**	222.1 ± 70.3 (206.8)	145.8 ± 99.2 (133.0)	0.002 *
**PNI (median)**	46.8 ± 6.2 (46.6)	48.3 ± 5.4 (47.3)	0.183
**Treatment interval change (%)**			
**ΔBW% ** (median)**	−4.1 ± 0.8 (−4.6)	−5.2 ± 1.1 (−6.3)	0.426
**ΔBMI% ** (median)**	−3.8 ± 0.8 (−4.7)	−5.4 ± 1.2 (−6.7)	0.235
**ΔHb% ** (median)**	−8.1 ± 1.8 (−9.5)	−11.1 ± 1.9 (−11.0)	0.163
**ΔWBC% ** (median)**	−21.7 ± 5.2 (−31.7)	−24.0 ± 5.9 (−34.6)	0.768
**ΔPlatelet% ** (median)**	−20.7 ± 3.7 (−30.1)	−13.9 ± 5.3 (−20.6)	0.304
**ΔTLC% ** (median)**	−39.2 ± 5.9 (−45.1)	−52.1 ± 4.7 (−51.7)	0.103
**ΔAlbumin% ** (median)**	6.7 ± 2.8 (5.0)	3.5 ± 2.5 (4.2)	0.448
**ΔCRP% ** (median)**	58.2 ± 7.8 (55.5)	47.6 ± 19.3 (68.6)	0.174
**ΔNLR% ** (median)**	4.7 ± 1.8 (2.4)	6.3 ± 1.4 (2.4)	0.514
**ΔPLR% ** (median)**	206.5 ± 23.7 (72.3)	218.6 ± 24.5 (129.9)	0.074
**ΔPNI% ** (median)**	−5.1 ± 1.9 (−4.9)	−10.5 ± 1.8 (−11.9)	0.053
**DXA-related measurements**			
**Before CCRT**			
**LBM (kg) (median)**	43.7 ± 5.1 (43.2)	43.6 ± 6.8 (43.4)	0.846
**TFM (kg) (median)**	17.0 ± 8.8 (15.4)	15.9 ± 6.0 (15.3)	0.450
**ASM (kg) (median)**	18.4 ± 3.1 (18.4)	18.7 ± 3.6 (19.0)	0.636
**Android (%) (median)**	29.6 ± 13.4 (29.5)	30.2 ± 10.5 (30.1)	0.814
**Gynoid (%) (median)**	25.7 ± 5.3 (25.5)	24.9 ± 6.1 (24.6)	0.551
**Treatment interval change (%)**			
**ΔLBM% ** (median)**	−6.1 ± 5.7 (−6.0)	−5.6 ± 6.6 (−5.1)	0.685
**ΔTFM% ** (median)**	−2.6 ± 16.5 (−4.0)	−5.8 ± 18.5 (−6.9)	0.308
**ΔASM% ** (median)**	−7.8 ± 7.7 (−8.4)	−8.0 ± 9.8 (−6.6)	0.901
**ΔAndroid% ** (median)**	−0.3 ± 21.6(−0.6)	−2.2 ± 28.9 (−2.3)	0.577
**ΔGynoid% ** (median)**	4.4 ± 14.6 (2.7)	3.3 ± 20.7 (1.9)	0.551
**Mean daily calorie intake during CCRT (kcal/kg/day, median)**	28.6 ± 8.6 (27.1)	26.6 ± 7.4 (25.7)	0.040 *
**<30 (%)**	69.6	81.5	0.131
**≥30 (%)**	30.4	18.5	
**Mean daily protein intake during CCRT (g/kg/day, median)**	1.1 ± 0.6 (1.2)	1.0 ± 0.9 (1.1)	0.324
**Feeding tube placement (%)**	68.1	37.3	0.022 *
**Mean days of feeding tube placement during CCRT (median)**	34.5 ± 3.5	20.5 ± 3.7	0.017 *
**CCRT data**			
**Radiotherapy**			
**Dose (Gy) (median)**	64.3 ± 4.5 (64.0)	69.7 ± 3.1 (69.3)	<0.001 *
**Fractions (median)**	31.9 ± 2.6 (32.0)	33.4 ± 1.5 (32.9)	<0.001 *
**Duration (days) (median)**	48.0 ± 4.8 (47)	51.6 ± 8.1 (50)	0.003 *
**Cisplatin dose (mg/m^2^) (median)**	238.4 ± 45.5 (240)	213.2 ± 66.0 (209)	0.009 *
**Grade 3/4 toxicity during CCRT**			
**Non-hematologic**			
**Dermatitis (%)**	4.3	3.7	0.858
**Pharyngitis (%)**	5.7	13.0	0.058
**Infection (%)**	14.4	31.5	0.024 *
**Mucositis (%)**	26.0	25.9	0.870
**Emesis (%)**	8.7	7.4	0.795
**Hematologic**			
**Anemia (%)**	7.2	11.0	0.289
**Neutropenia (%)**	33.3	38.9	0.524
**Thrombocytopenia**	5.7	13.0	0.167
**2-year mortality rate (%)**	43.5	29.6	0.115

* Compare the value difference between the oral cavity and non-oral cavity cancers for each variable (*p* < 0.05). Abbreviations: OCC, oral cavity cancer; NOCC, non-oral cavity cancer; CCRT, concurrent chemoradiotherapy; SD, standard deviation; HN-CCI, Charlson Comorbidity Index; ECOG, Eastern Cooperative Oncology Group; PG-SGA, Patient-Generated Subjective Global Assessment; BW, body weight; BMI, body mass index; Hb, hemoglobin; WBC, white blood cell; TLC, total lymphocyte count; CRP, C-reactive protein; NLR, neutrophil-to-lymphocyte ratio; PLR, platelet-to-lymphocyte ratio; PNI, prognostic nutritional index; DXA, dual-energy X-ray absorptiometry; LBM, lean body mass; TFM, total fat mass; ASM, appendicular skeletal mass. **** Δ** indicates a value obtained by subtracting the pretreatment value from the posttreatment value. % indicates (**Δ** value/ the pretreatment value) × 100%.

**Table 2 biomedicines-10-00388-t002:** Baseline characteristics and treatment interval changes during the CCRT course in 123 patients with LAHNSCC undergoing CCRT stratified by tumor location and CCRT settings stratified by 30 kcal/kg/day calorie supply.

	OCC with Adjuvant CCRT			NOCC with Primary CCRT		
	CCRT Completion			CCRT Completion		
Variables, Expressed as Numbers (%) or Mean ± SD	25–30 kcal/kg/day	≥30 kcal/kg/day	*p*-Value	25–30 kcal/kg/day	≥30 kcal/kg/day	*p*-Value *
*Patient number*	48 (69.6)	21 (30.4)		43 (79.6)	11 (20.4)	
*Clinicopathologic*						
**Age (years)**	52.6 ± 8.7	54.5 ± 7.9	0.389	55.3 ± 8.5	55.2 ± 8.9	0.961
**Sex (male/female)**	47 (97.9): 1 (2.1)	21 (100.0): 0 (0.0)	0.505	41 (95.3): 2 (4.7)	10 (90.9): 1 (9.1)	0.566
**Tumor location (OCC/NOCC)**			0.353			0.441
**Buccal mucosa/Tonsil**	17 (35.4)	3 (14.2)		10 (23.2)	3 (27.2)	
**Tongue/Tongue base**	18 (37.5)	10 (47.6)		5 (11.6)	1 (9.1)	
**Gingiva/Soft palate**	8 (16.6)	5 (23.8)		2 (4.7)	1 (9.1)	
**Mouth floor/Hypopharynx**	2 (4.2)	1 (4.8)		19 (44.2)	5 (45.5)	
**Retromolar/Larynx**	1 (2.1)	1 (4.8)		7 (16.3)	1 (9.1)	
**Lip/Nasopharynx**	2 (4.2)	0 (0.0)		--	--	
**Hard palate**	0 (0.0)	1 (4.8)		--	--	
**TNM stage (III vs. IVA vs. IVB)**	2 (4.2): 34 (70.8): 12 (25.0)	2 (9.5): 32 (76.2): 3 (14.3)	0.460	4 (9.3): 28 (65.1): 11 (25.6)	1 (9.1): 7 (63.6): 3 (27.3)	0.993
**T status (T0-2 vs. T3-4)**	6 (12.5): 42 (87.5)	2 (9.5): 15 (90.5)	0.722	17 (39.5): 26 (60.5)	3 (27.2): 8 (72.8)	0.452
**N status (N0-1 vs. N2-3)**	22 (45.8): 26 (54.2)	8 (38.1):13 (61.9)	0.551	8 (18.6): 35 (81.4)	3 (27.2): 8 (72.8)	0.524
**ECOG performance status (0: 1: 2)**	2 (4.2): 43 (89.5): 3 (6.3)	0 (0.0): 18 (85.7): 3 (14.3)	0.371	6 (14.0): 37 (86.0): 0 (0.0)	0 (0.0): 10 (90.9): 1 (9.1)	0.065
**Histological grade (1: 2: 3)**	6 (12.5): 35 (72.9): 7 (14.6)	2 (9.5): 16 (76.2): 3 (14.3)	0.935	2 (4.4): 29 (63.0): 15 (32.6)	1 (8.3): 6 (50.0): 5 (41.7)	0.675
**Smoking (no/yes)**	5 (10.4): 43 (89.6)	1 (4.5): 20 (90.2)	0.443	4 (9.3): 39 (90.7)	1 (9.1): 10 (90.9)	0.983
**Alcohol (no/yes)**	13 (27.1): 351 (72.9)	5 (23.8): 16 (76.2)	0.776	12 (27.9): 31 (72.1)	1 (9.1): 10 (90.9)	0.193
**Betel nut (no/yes)**	11 (22.9): 37 (77.1)	5 (23.8): 16 (76.2)	0.936	20 (46.5): 23 (53.5)	7 (68.6): 4 (31.4)	0.311
**HN-CCI (0 vs. 1 vs. 2 vs. ≥3)**	19 (39.5): 13 (27.1): 2 (6.3): 13 (27.1)	10 (47.6): 2 (9.5): 3 (14.3): 6 (28.1)	0.478	16 (37.1): 12 (27.9): 6 (14.0): 9 (21.0)	5 (45.5): 4 (36.4): 0 (0.0): 2 (18.1)	0.715
**Tracheostomy (no/yes)**	19 (39.6): 29 (60.4)	4 (19.0): 17 (81.0)	0.096	22 (47.8): 24 (52.2)	7 (68.6): 4 (31.4)	0.088
**PG-SGA (stage A vs. stage B vs. stage** **C) before CCRT**	8 (16.7): 29 (60.4): 11 (22.9)	5 (23.8): 9 (42.9): 7 (33.3)	0.402	3 (8.7): 28 (63.0): 12 (28.3)	2 (18.2): 5 (45.4): 4 (36.4)	0.377
*Anthropometric and blood NIB data*						
*Before CCRT*						
**BW (kg)**	65.3 ± 12.5	54.9 ± 7.7	<0.001 *	65.1 ± 10.5	51.0 ± 11.7	<0.001 *
**BMI (kg/m^2^)**	23.9 ± 0.4	20.2 ± 0.7	0.001 *	23.6 ± 3.5	19.1 ± 3.8	0.001 *
**Hb (g/dL)**	11.7 ± 1.6	11.6 ± 1.0	0.780	11.9 ± 1.7	11.8 ± 1.6	0.820
**WBC (×10^3^ cells/mm^3^)**	7.1 ± 2.4	7.6 ± 2.7	0.409	7.4 ± 3.2	5.9 ± 1.8	0.144
**Platelet count (×10^3^/mm^3^)**	324.6 ± 126.4	378.9 ± 186.2	0.233	246.4 ± 71.6	270.0 ± 90.0	0.363
**TLC (×10^3^ cells/mm^3^)**	1.0 ± 0.6	1.1 ± 0.7	0.905	1.9 ± 0.7	1.6 ± 0.4	0.115
**Albumin (g/dL)**	3.8 ± 0.5	3.8 ± 1.5	0.601	3.9 ± 0.4	3.6 ± 0.6	0.055
**CRP (mg/dL)**	10.5 ± 1.9	13.0 ± 4.0	0.521	18.1 ± 7.2	20.6 ± 12.6	0.879
**NLR**	3.4 ± 0.4	5.3 ± 2.5	0.307	2.9 ± 0.5	2.7 ± 0.4	0.809
**PLR**	216.4 ± 23.7	235.1 ± 31.4	0.678	137.1 ± 28.9	158.4 ± 23.0	0.643
**PNI**	47.1 ± 5.9	46.3 ± 6.8	0.661	48.9 ± 5.2	46.0 ± 5.7	0.073
*Treatment interval change (%)*						
**ΔBW% ****	−5.3 ± 0.9	−1.4 ± 1.1	0.019 *	−6.4 ± 1.2	−0.2 ± 2.1	0.023 *
**ΔBMI% ****	−4.9 ± 0.9	−1.3 ± 1.2	0.040 *	−6.7 ± 1.2	−0.4 ± 2.0	0.001 *
**ΔHb% ****	−7.4 ± 2.2	−9.7 ± 2.7	0.538	−12.9 ± 2.0	−7.2 ± 4.8	0.225
**ΔWBC% ****	−24.3 ± 4.6	−15.5 ± 13.6	0.445	−27.8 ± 6.2	−13.0 ± 14.1	0.198
**ΔPlatelet% ****	−20.1 ± 4.4	−21.9 ± 6.8	0.818	−17.8 ± 6.1	−29.3 ± 7.9	0.152
**ΔTLC% ****	−41.5 ± 4.9	−33.7 ± 16.1	0.550	−49.2 ± 5.5	−63.2 ± 8.0	0.238
**ΔAlbumin% ****	6.3 ± 2.7	7.6 ± 7.8	0.847	1.4 ± 2.8	1.9 ± 8.2	0.237
**ΔCRP% ****	84.5 ± 10.8	98.0 ± 6.6	0.614	55.1 ± 22.3	80.5 ± 42.1	0.404
**ΔNLR% ****	5.3 ± 2.5	3.4 ± 1.8	0.027 *	4.8 ± 7.1	12.4 ± 23.6	0.292
**ΔPLR% ****	113.9 ± 30.9	89.8 ± 33.1	0.684	137.1 ± 27.8	237.2 ± 78.8	0.185
**ΔPNI% ****	−4.8 ± 2.3	−5.8 ± 3.3	0.807	−9.9 ± 1.8	−11.6 ± 4.5	0.692
*Body composition parameters*						
*Before CCRT*						
**LBM (kg)**	45.2 ± 5.1	40.4 ± 3.6	<0.001 *	44.8 ± 6.2	38.4 ± 7.2	0.004 *
**TFM (kg)**	19.3 ± 9.1	11.8 ± 5.4	0.001 *	17.4 ± 5.5	10.2 ± 4.8	<0.001 *
**ASM (kg)**	19.2 ± 2.9	16.6 ± 2.5	0.001 *	19.6 ± 3.4	15.4 ± 3.8	0.001 *
**Android (%)**	32.8 ± 12.9	22.3 ± 11.5	0.002 *	32.7 ± 9.1	19.9 ± 9.6	<0.001 *
**Gynoid (%)**	27.4 ± 8.2	21.7 ± 7.2	0.007 *	26.4 ± 9.7	19.4 ± 4.4	<0.001 *
*Treatment interval change (%)*						
**ΔLBM% ****	−7.1 ± 5.8	−3.9 ± 4.7	0.001 *	−5.7 ± 7.2	−2.4 ± 4.0	0.021 *
**ΔTFM% ****	−7.5 ± 13.9	8.1 ± 17.2	<0.001 *	−8.9 ± 16.0	6.1 ± 6.3	0.015 *
**ΔASM% ****	−8.4 ± 7.5	−6.3 ± 8.0	0.280	−8.7 ± 10.3	−5.1 ± 7.1	0.276
**ΔAndroid% ****	−4.8 ± 19.2	11.9 ± 22.5	0.002 *	−7.8 ± 5.2	19.2 ± 12.8	0.005 *
**ΔGynoid% ****	2.6 ± 12.2	8.5 ± 18.7	0.126	1.0 ± 2.6	10.9 ± 8.5	0.109
*Mean daily calorie intake during CCRT*	25.2 ± 4.3	38.6 ± 7.0	<0.001 *	25.7 ± 3.8	36.6 ± 7.8	<0.001 *
*Mean protein intake during CCRT*	0.9 ± 0.9	1.3 ± 0.4	0.014 *	0.8 ± 0.9	1.2 ± 0.8	0.021 *
*Feeding tube placement (no/yes)*	18 (37.5): 30 (63.5)	4 (19.0): 17 (81.0)	0.130	21 (48.8): 22 (51.2)	5 (45.5): 6 (54.5)	0.841
*Mean days of feeding tube placement*	28.2 ± 4.1	48.8 ± 5.7	0.007 *	20.6 ± 4.0	27.5 ± 9.5	0.451
*CCRT data*						
*Radiotherapy*						
**Dose (Gy)**	64.3 ± 4.5	64.3 ± 1.7	0.988	69.8 ± 2.9	69.7 ± 4.1	0.911
**Fractions**	31.8 ± 1.7	32.3 ± 1.6	0.299	33.3 ± 1.1	34.0 ± 2.3	0.164
**Duration (days)**	45.2 ± 5.4	47.7 ± 3.6	0.727	51.9 ± 8.5	50.3 ± 6.1	0.538
*Cisplatin dose (mg/m^2^)*	238.2 ± 23.3	239.1 ± 16.5	0.880	211.3 ± 10.2	220.9 ± 12.7	0.670
*Grade 3/4 toxicity during CCRT*						
**Non-hematologic**						
**Dermatitis (no/yes)**	46 (95.8): 2 (4.2)	20 (95.2): 1 (4.8)	0.911	41 (95.7): 2 (4.3)	11 (100.0): 0 (0.0)	0.466
**Pharyngitis (no/yes)**	45 (93.8): 3 (6.2)	19 (90.5): 2 (9.5)	0.629	36 (83.7): 7 (16.3)	8 (72.8): 3 (27.2)	0.402
**Infection (no/yes)**	42 (87.5): 6 (12.5)	17 (81.0): 4 (19.0)	0.477	32 (74.4): 11 (25.6)	5 (45.5): 6 (54.5)	0.065
**Mucositis (no/yes)**	35 (72.9): 13 (27.1)	17 (81.0): 4 (19.0)	0.476	32 (74.4): 11 (25.6)	8 (72.8): 3 (27.2)	0.909
**Emesis (no/yes)**	43 (89.6): 5 (10.4)	20 (95.2): 1 (4.8)	0.476	40 (93.0): 3 (7.0)	10 (90.9): 1 (9.1)	0.811
**Hematologic**						
**Anemia (no/yes)**	44 (91.7): 4 (8.3)	20 (95.2): 1 (4.8)	0.599	37 (86.0): 6 (14.0)	7 (90.9): 4 (9.1)	0.668
**Neutropenia (no/yes)**	31 (64.6): 17 (35.4)	15 (71.4): 6 (28.6)	0.579	26 (60.5): 17 (39.5)	8 (63.6): 4 (36.4)	0.847
**Thrombocytopenia (no/yes)**	45 (93.8): 3 (6.2)	20 (95.2): 1 (4.8)	0.808	37 (86.0): 6 (14.0)	10 (90.9): 1 (9.1)	0.668
2-year mortality (%)	18 (37.1)	12 (57.1)	0.045 *	10 (23.3)	6 (54.5)	0.043 *

* indicates a significant *p*-value < 0.05. The analytical methods used with data presented in Appendix A are identical to those used with data presented in Table 3. Abbreviations: LAHNSCC, locally advanced head and neck squamous cell carcinoma; CCRT, concurrent chemoradiotherapy; OCC, oral cavity cancer; NOCC, non-oral cavity cancer; TNM, tumor node metastasis; ECOG PS, Eastern Collaboration Oncology Group performance status; HN-CCI, head and neck Charlson Comorbidity Index; RT, radiotherapy; PG-SGA, Patient-Generated Subjective Global Assessment; BMI, body mass index; BW, body weight; Hb, hemoglobin; WBC, white blood cell count; TLC, total lymphocyte count; CRP, C-reactive protein; LBM, lean body mass; TFM, total fat mass; ASM, appendicular skeletal muscle mass; BMC, bone mineral content. Independent *t*-tests were used for age, BW, BMI, Hb, TLC, and albumin levels. The Mann–Whitney test was used for WBC, platelet count, CRP, and all body composition parameters. The chi-square test was used for categorical data. **** Δ** indicates a value obtained by subtracting the pretreatment value from the post-treatment value. % indicates (**Δ** value/pretreatment value) × 100%.

**Table 3 biomedicines-10-00388-t003:** Factor analysis results of BW, BMI, and DXA-related parameters of 69 patients with OCC treated with postoperative adjuvant CCRT and 54 patients with NOCC treated with primary CCRT.

	OCC with Postoperative Adjuvant CCRT				NOCC with Primary CCRT			
	Factor				Factor			
Component	1	2	3	4	1	2	3	4
**BW**	0.766	−0.103	−0.092	0.614	0.834	0.500	−0.195	−0.054
**BMI**	0.804	−0.115	−0.082	0.508	0.656	0.656	−0.268	−0.016
**LBM**	0.265	−0.110	0.102	0.931	0.965	0.127	−0.193	−0.007
**ASM**	0.287	−0.121	0.867	0.897	0.956	0.180	−0.175	−0.048
**TFM**	0.930	−0.084	−0.337	0.302	0.514	0.827	−0.162	−0.056
**Android**	0.937	−0.072	−0.093	0.155	0.353	0.863	−0.219	−0.080
**Gynoid**	0.964	−0.052	−0.061	−0.032	0.032	0.935	−0.064	−0.155
**ΔBW%**	0.005	0.460	0.732	−0.348	−0.185	−0.267	0.790	0.390
**ΔBMI%**	0.043	0.400	0.717	−0.341	−0.134	−0.305	0.794	0.362
**ΔLBM%**	−0.072	0.031	0.923	−0.046	−0.223	−0.062	0.886	−0.239
**ΔASM%**	−0.086	0.137	0.887	0.042	−0.134	−0.048	0.853	−0.014
**ΔTFM%**	−0.133	0.900	0.322	−0.172	−0.063	−0.105	0.200	0.951
**ΔAndroid%**	−0.193	0.895	−0.150	−0.081	−0.089	−0.125	0.182	0.907
**ΔGynoid%**	0.019	0.936	−0.049	0.005	0.076	−0.016	−0.222	0.942
**Eigenvalue**	4.51	2.21	1.27	1.06	4.82	2.04	1.42	1.05
**% of accumulative variances**	45.1	67.2	79.9	90.5	48.2	68.6	82.8	93.8

BW, body weight; BMI, body mass index; DXA, dual-energy X-ray absorptiometry; OCC, oral cavity cancer; NOCC, non-oral cavity cancer; CCRT, concurrent chemoradiotherapy; LBM, lean body mass; TFM, total fat mass; ASM, appendicular skeletal muscle. **Δ** indicates a value obtained by subtracting the pretreatment value from the post-treatment value. % indicates (**Δ** value/pretreatment value) × 100%.

**Table 4 biomedicines-10-00388-t004:** Univariate and multivariate Cox regression analyses of factors associated with 2-year mortality rates of 69 OCC patients treated with postoperative adjuvant CCRT and 54 NOCC patients treated with primary CCRT.

	OCC with Postoperative Adjuvant CCRT			NOCC with Primary CCRT		
Variables	Univariate	Multivariate		Univariate	Multivariate	
	*p*-Value	HR (95% CI)	*p*-Value	*p*-Value	HR (95% CI)	*p*-Value
** *Clinicopathologic* **						
Sex (ref: female)	0.979			0.860		
Age	0.949			0.014 *	1.088 (1.019~1.093)	0.012 *
TNM stage (ref: IV)	0.982			0.211		
T status (ref: T3-4)	0.347			0.166		
N status (ref: N2-3)	0.161			0.156		
Histologic grade (ref: poorly differentiated)	0.019 *			0.810		
ECOG performance status (ref: 2)	0.012 *	0.180 (0.038~0.842)	0.006 *	0.333		
Smoking (ref: yes)	0.645			0.487		
Alcohol (ref: yes)	0.053			0.440		
Betel nut (ref: yes)	0.264			0.996		
HN-CCI (ref: no)	0.047 *			0.255		
Tracheostomy (ref: yes)	0.732			0.055		
** *CCRT* **						
RT dose (Gy)	0.446			0.222		
RT fractions	0.255			0.411		
RT duration (days)	0.140			0.101		
Cisplatin dose	0.774			0.458		
** *CCRT-induced grade 3/4 toxicity* **						
Dermatitis (ref: yes)	0.037 *			0.746		
Pharyngitis (ref: yes)	0.012 *			0.237		
Mucositis (ref: yes)	0.958			0.885		
Infection (ref: yes)	0.394			0.027 *		
Emesis (ref: yes)	0.218			0.469		
Anemia (ref: yes)	0.970			0.211		
Neutropenia (ref: yes)	0.855			0.969		
Thrombocytopenia (ref: yes)	0.306			0.555		
** *Mean daily calorie intake during CCRT (ref: ≥30)* **	0.045 *			0.039 *		
** *Mean protein intake during CCRT* **	0.269			0.478		
** *Feeding tube placement (ref: yes)* **	0.039 *			0.052		
** *Mean days of feeding tube placement during CCRT* **	0.001 *			0.022 *		
** *Nutritional and inflammatory markers* **						
**Before CCRT**						
BMI (kg/m^2^)	0.005 *			0.011 *		
BW (kg)	0.004 *			0.008 *		
Hb (g/dL)	0.128			0.098		
WBC (×10^3^ cells/mm^3^)	0.061			0.621		
Platelet count (×10^3^/mm^3^)	0.024 *			0.564		
TLC (×10^3^ cells/mm^3^)	0.133			0.101		
Albumin (g/dL)	0.191			0.056		
CRP (mg/dL)	0.038 *			0.541		
NLR	0.179			0.023 *	1.250 (1.014~1.311)	0.027 *
PLR	<0.001 *	1.004 (1.002~1.005)	<0.001 *	0.111		
PNI	0.022 *			0.009 *		
**Treatment interval changes (%)**						
ΔBW% **	0.035 *			0.502		
ΔBMI% **	0.121			0.440		
ΔHb% **	0.544			0.052		
ΔWBC% **	0.818			0.008 *		
ΔPlatelet% **	0.041 *			0.543		
ΔTLC% **	0.481			0.798		
ΔAlbumin% **	0.346			0.410		
ΔCRP% **	0.046 *			0.055		
ΔNLR% **	0.009 *			0.048 *		
ΔPLR% **	0.733			0.122		
ΔPNI% **	0.419			0.869		
** *PG-SGA before CCRT (ref: stage C)* **	0.007 *			0.098		
** *Body composition parameters* **						
Factor 1	0.023 *			0.430		
Factor 2	0.871			0.008 *	0.477 (0.020~0.777)	0.008 *
Factor 3	0.003 *	1.562 (1.094~2.229)	0.014 *	0.054		
Factor 4	0.098			0.358		

* indicates a significant *p*-value < 0.05 Abbreviations: CCRT, concurrent chemoradiotherapy; HR, hazard ratio; CI, confidence interval; TNM, tumor node metastasis; ECOG, Eastern Collaboration Oncology Group; HN-CCI, head and neck Charlson Comorbidity Index; RT, radiotherapy; PG-SGA, Patient-Generated Subjective Global Assessment; BMI, body mass index; BWL, body weight loss; Hb, hemoglobin; WBC, white cell count; TLC, total lymphocyte count; CRP, C-reactive protein; NLR, neutrophil-to-lymphocyte ratio; PLR, platelet-to lymphocyte ratio; PNI, prognostic nutritional index. **** Δ** indicates a value obtained by subtracting the pretreatment value from the post-treatment value. % indicates (**Δ** value/the pretreatment value) × 100%.

## Data Availability

The data presented in this study are available on request from the corresponding author.

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
