# Peer review of "Inflammation Status and Body Composition Predict Two-Year Mortality of Patients with Locally Advanced Head and Neck Squamous Cell Carcinoma under Provision of Recommended Energy Intake during Concurrent Chemoradiotherapy"

_biomedicines, 2022, doi:10.3390/biomedicines10020388_

Round 1

Reviewer 1 Report

Inflammation Status and Body Composition Predict Two-Year Mortality of Patients with Locally Advanced Head and Neck Squamous Cell Carcinoma under Provision of Recommended Energy Intake during Concurrent Chemoradiotherapy

This prospective observational study examines a very interesting topic, since most patients with locally advanced head and neck squamous cell carcinoma require concurrent chemoradiotherapy to improve disease control. The aim of this work is to investigate the impact of recommended calorie intake, nutrition-inflammation biomarkers and total body composition change (assessed using dual-energy X-ray absorptiometry), on patients’ clinical benefits, such as lower treatment-associated toxicity and better survival outcomes.

The article is interesting and takes into account many crucial issues about the management of patients affected by LAHNSCC, as the above-mentioned conditions can lead to delayed treatment, poor quality of life and -at worst- unexpected mortality.

Patients with LAHNSCC who consumed at least 25 kcal/kg/day during CCRT were prospectively recruited and stratified into two subgroups, since LAHNSCC is a heterogeneous disease with different patient characteristics and therapeutic intent: OCC with postoperative adjuvant CCRT and NOCC with primary curative intent CCRT.

One of the points of strength of this article is the detailed materials and methods section, in which the authors describe very clearly how the work was structured. The statistical analysis performed is accurate and well-structured.

The authors also provided a very detailed discussion, and their assumptions seem to be well-grounded, with an accurate and interesting literature analysis.

The authors are aware that this research has several limitations, as underlined in the discussion section. The first is that the enrolled participants were Taiwanese, and most were men (more than 97 %), so the current findings may not be applicable to non-Taiwanese patients. Another limit to highlight is the lack of appropriate adjustment for protein intake, including the quantity and quality of protein.

The current work has gone some way towards enhancing the understanding of the role of patients’ characteristics, inflammation status and body composition in the 2-year mortality of LAHNSCC patients; it would be interesting to broaden this work to investigate whether increased protein intake or modulation of protein quality could affect treatment outcomes and the applicability of these results to non-Taiwanese patients.

The paper is suitable for publication.

Author Response

 We sincerely thank the reviewer for the interest in our patient cohort and results.

Reviewer 2 Report

Manuscript entitled "Inflammation Status and Body Composition Predict Two-Year Mortality of Patients with Locally Advanced Head and Neck Squamous Cell Carcinoma under Provision of Recommended Energy Intake during Concurrent Chemoradiotherapy"

My oppions are as follows:   1. This work is a clinical study in which the data presented and the main theme are the same with that one recently published in  Nutrients by the authors. It is not ideal. https://www.mdpi.com/2072-6643/13/9/2969

2. The classification of tumor should be modified. Since this study are specific for Head and Neck Squamous Cell Carcinoma, it is not ideal to include cancer from nasopharynx. Because the can biology of NP cancer is totally different.

3. The clinicopathologic factors should be described in more detail. For example, the deapth of tumor imvasion, the presence of vascular and perineurial invasion, and p16 staining status.

Round 2

Reviewer 2 Report

I suggest the authors withdraw this work to protect themselves, since there are ethic issues.

Round 3

Reviewer 2 Report

no comment.
